

# Choice-based severity scale (CSS): assessing the relative severity of procedures from a laboratory animal's perspective

Lauren Cassidy[1,2], Stefan Treue[1,3,4], Alexander Gail[1,3,4] and Dana Pfefferle[1,3]

[1] Welfare and Cognition Group, Cognitive Neuroscience Lab, German Primate Center, Goettingen, Lower Saxony, Germany
[2] Population and Behavioral Health Services, California National Primate Research Center, University of California, Davis, California, United States
[3] Leibniz-ScienceCampus Primate Cognition, Goettingen, Lower Saxony, Germany
[4] Faculty for Biology and Psychology, University of Goettingen, Goettingen, Lower Saxony, Germany

Corresponding author
Lauren Cassidy,
lccassidy@ucdavis.edu

## ABSTRACT

One primary goal of laboratory animal welfare science is to provide a comprehensive severity assessment of the experimental and husbandry procedures or conditions these animals experience. The severity, or degree of suffering, of these conditions experienced by animals are typically scored based on anthropocentric assumptions. We propose to (a) assess an animal's subjective experience of condition severity, and (b) not only rank but scale different conditions in relation to one another using choice-based preference testing. The Choice-based Severity Scale (CSS) utilizes animals' relative preferences for different conditions, which are compared by how much reward is needed to outweigh the perceived severity of a given condition. Thus, this animal-centric approach provides a common scale for condition severity based on the animal's perspective. To assess and test the CSS concept, we offered three opportunistically selected male rhesus macaques (*Macaca mulatta*) choices between two conditions: performing a cognitive task in a typical neuroscience laboratory setup (laboratory condition) *versus* the monkey's home environment (cage condition). Our data show a shift in one individual's preference for the cage condition to the laboratory condition when we changed the type of reward provided in the task. Two additional monkeys strongly preferred the cage condition over the laboratory condition, irrespective of reward amount and type. We tested the CSS concept further by showing that monkeys' choices between tasks varying in trial duration can be influenced by the amount of reward provided. Altogether, the CSS concept is built upon laboratory animals' subjective experiences and has the potential to de-anthropomorphize severity assessments, refine experimental protocols, and provide a common framework to assess animal welfare across different domains.

## INTRODUCTION

Animal research models (*i.e.*, laboratory animals) are crucial for advancing scientific knowledge across many fields (*e.g.*, *Kiros et al., 2012*; *Roelfsema & Treue, 2014*; *Bale et al., 2019*; *Meyerholz, Beck & Singh, 2020*; *Homberg et al., 2021*; *Azkona & Sanchez-Pernaute, 2022*). Good animal welfare is not only important for the health and well-being of these animals but also to the quality and validity of the research in which they are involved (*Poole, 2007*; *Jennings & Prescott, 2009*). Therefore, it is our duty as researchers and caretakers of laboratory animals to ensure their care and welfare meets a high standard. However, animals cannot naturally linguistically report how they are experiencing different experimental and husbandry events. Caretakers and researchers must instead indirectly infer animal welfare by changes and/or differences in their physiology, natural behavior, and psychology in relation to these events. So how is animal welfare currently measured?

Welfare and severity assessments have been developed to quantify and understand the impact that research has on laboratory animals (*e.g.*, Extended Welfare Assessment Grid: *Honess & Wolfensohn, 2010*; *Wolfensohn et al., 2015*; Qualitative Behavioural Assessment: *Wemelsfelder, 2007*; score sheets: *Bugnon, Heimann & Thallmair, 2016*; *Ullmann et al., 2018*). In some assessments, different welfare parameters are nested within overarching domains (*e.g.*, physical, psychological, procedural, environmental), which are broken down into the different putative conditions (*e.g.*, procedures, events, states) that can be experienced. For example, social housing, a common welfare parameter for non-human primates, would fall under the environmental domain and could consist of four different conditions: group-housing, continuous pair-housing, intermittent pair-housing, and single housing (*Hannibal et al., 2017*). Each housing condition is comprised of different elements such as the number of social partners available, the duration and/or extent that physical contact with a social partner is possible, and the amount of available cage space. The state of these elements can differ between conditions; for example, the extent that physical contact with a social partner is possible is full-time in the continuous pair-housing condition and non-existent in the single housing condition. Based on these differences, conditions are ranked in relation to one another and given a score based on their putative impact on welfare as assessed by humans.

During a severity assessment, a given welfare parameter is quantified (*i.e.*, scored) based on the current condition that the animal is experiencing. Generally, these scores are combined to create a composite score for the domain. While this type of severity assessment provides a great overview of what the animal experiences over the course of its life, the hierarchies of the conditions within some welfare parameters are still determined by anthropocentric judgments. These judgments are prone to observer and confirmatory biases that may not reflect an individual's actual experience as they likely experience procedures differently (*Tuyttens et al., 2014*; *Bello et al., 2014*). Presently, scores given to welfare parameters are also assumed to be comparable within and across domains. However, it is unknown whether, for example, the highest score of a welfare parameter in the experimental domain (*e.g.*, performing a task in a laboratory setup) is equivalent to the

highest score of a welfare parameter in the environmental domain (*e.g.*, single housing). Such comparisons are difficult as welfare parameters differ in their function and conditions are often comprised of elements that have different associated costs and benefits (*i.e.*, 'comparing apples with oranges'). It may also be that the different domains are not orthogonal as is often assumed as there may be dependencies between welfare parameters. For instance, the weight of an animal (*e.g.*, clinical status) likely correlates with its daily activity (*e.g.*, behavior).

Determining animals' preferences can reveal how valuable certain resources are in relation to one another (*Hosey, Jacques & Burton, 1999*; *Kahnau et al., 2022*). Often preference tests are conducted by presenting an animal with a series of binary choices among an array of options to see how frequently each option is selected in relation to the others (*Habedank et al., 2018*). Preference testing becomes more challenging when the options are more complex and/or abstract (compared to, *e.g.*, choosing between favored foods or fluids: *Huskisson et al., 2020*; *Hansell, Åsberg & Laska, 2020*) as the decider, the animal, must weigh the combined costs and benefits of each. In such multi-faceted options, multiple decision variables are evaluated to optimize reward and effort and combined into a single value, the utility, which characterizes the desirability of each choice (see utility theory: *Von Neumann & Morgenstern, 1944*). For example, animals have preferences for tasks varying in difficulty and respond accordingly when the reward and/or effort contingencies are adjusted (*Suzuki & Matsuzawa, 1997*; *Calapai et al., 2017*). Outside of experimental tasks, animals have exhibited preferences for more complex options with respects to positive reinforcement training (*e.g.*, *Schapiro & Lambeth, 2007*), environmental parameters (*e.g.*, supplementary light: *Buchanan-Smith & Badihi, 2012*), enrichment (*e.g.*, *Hobbiesiefken et al., 2021*), and even determining the type and/or whether to cooperate with medical treatment (*e.g.*, *Magden et al., 2013*, *2016*; *Webb, Hau & Schapiro, 2018*). Previous work has advocated for the use of preference testing to guide animal welfare assessment, particularly for determining the value of different environment-based items to animals (*Habedank et al., 2018*; *Kahnau et al., 2020*, *2022*). Presently offering laboratory animals' choices between other welfare conditions, such as experimental procedures and husbandry practices, has not been conducted to our knowledge.

To complement existing welfare and severity assessments, we propose the Choice-based Severity Scale (CSS), a novel concept to improve how welfare parameters are measured in laboratory animals (see Fig. 1). By using choice tests, we can determine which one of two conditions within a welfare parameter is preferred by an animal, thus reflecting how it perceives the relative severity of these conditions. Hence, preferences can be used to rank conditions within a welfare parameter as having the lowest (most preferred condition) to highest (least preferred condition) impact on the well-being of laboratory animals. Since individuals likely differ in how strongly they prefer one condition over the other, we propose that preference strength can be determined by how much is needed to "pay" the animal to choose each condition by adjusting the reward parameters (amount and/or type of fluid) experienced in association with each condition. With this information, the scales of welfare parameters can be individualized, where the difference in the reward parameters

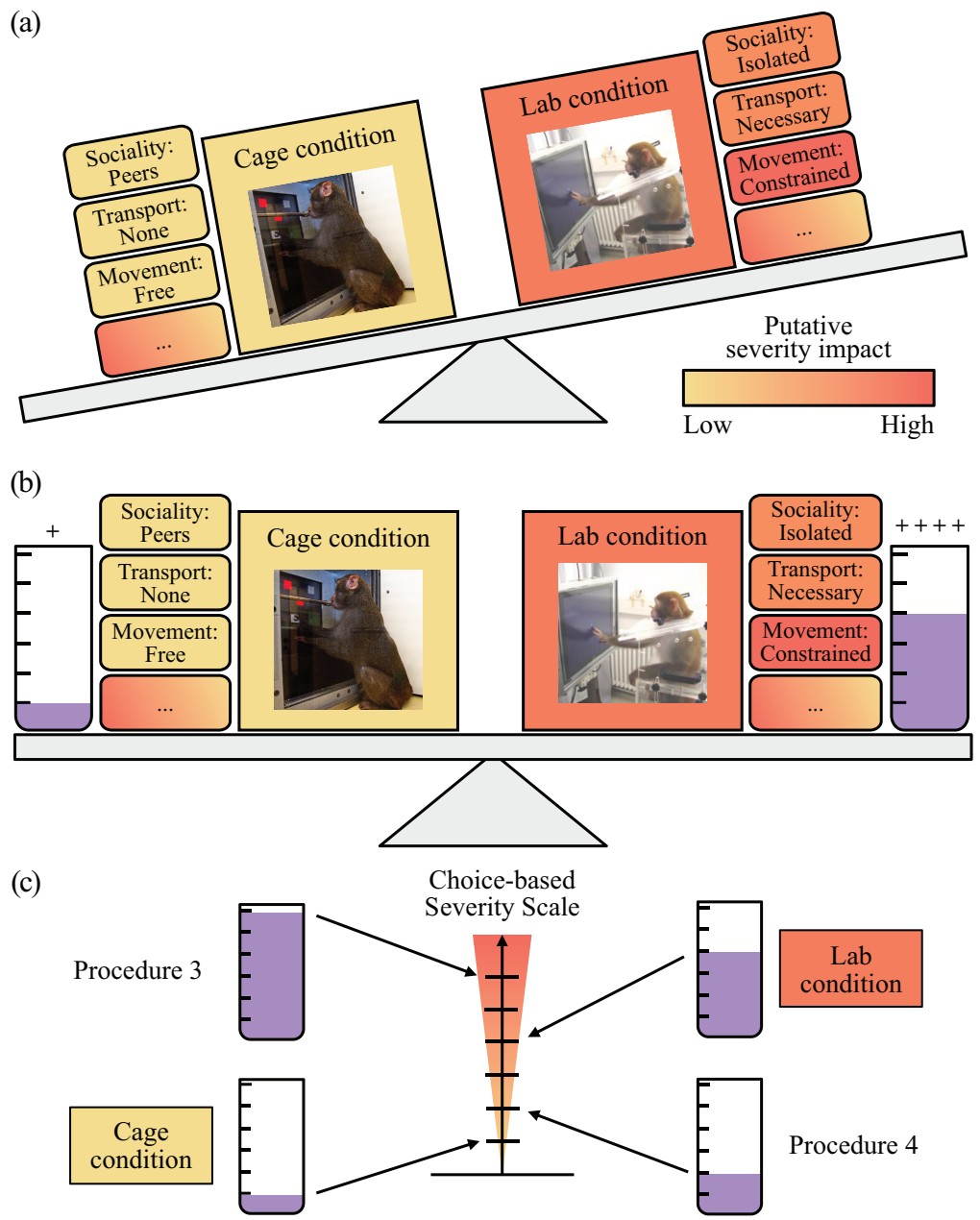

**Figure 1 Choice-based Severity Scale: A welfare assessment concept using choice-based preference testing in laboratory animals.** Laboratory is abbreviated as "Lab". (A) An example of two conditions, here experimental procedures, and their associated putative welfare costs and benefits (elements) positioned adjacently. Collectively the elements of the putatively less desirable condition, performing a task in a laboratory setup (*i.e.*, laboratory condition), have a higher severity impact than those of the preferred condition, performing a task close to the home cage (*i.e.*, cage condition). The ellipse indicates that there may be more elements to these conditions than those we have visualized. (B) By providing the animals substantially more reward to choose the less desirable condition, we can balance the experienced severity of the two conditions in relation to one another. (C) The amount of reward needed to pay the animals to choose each condition can be used a way to objectively rank and scale several conditions in relation to one another on a severity scale. Alexander Gail photographed the image of a monkey using the touchscreen in the laboratory condition. Ingo Bulla photographed the image of a monkey using a cognitive testing system close to their home cage.

**Table 1 The elements (location, transport, movement, sociality) expected to differ for each condition tested and their putative severity impact.**

| Condition | Elements | | | | Severity impact |
| | Location | Transport | Movement | Sociality | |
|---|---|---|---|---|---|
| Upper cage | Upper cage of testing compartment, adjacent to home cage | No | Free to move within the limits of upper cage | Visual, auditory, olfactory, but no tactile contact to conspecifics | * |
| Lower cage | Lower cage of testing compartment, adjacent to home cage | No | Free to move within the limits of lower cage | Auditory, olfactory, but no visual and tactile contact to conspecifics | ** |
| Lab | Neuroscience setup in isolated room | Yes | In non-human primate chair | No contact to other conspecifics (isolated) | *** |

Note:
Laboratory is abbreviated as "Lab". A greater number of asterisks indicates a higher putative severity impact to nonhuman primate welfare.

would serve to relatively rank and scale conditions within a welfare parameter based on the animal's perspective.

Once welfare parameters have been scaled using the same metric, comparing scores between different parameters becomes meaningful. Even combining scores given to welfare parameters from different domains into a single overall welfare score (as is commonly done in score sheets) becomes appropriate. For example, a score given to the housing parameter could now be combined with the scores determined for bedding type (*i.e.*, same domain as housing) and even cognitive testing location (*i.e.*, a different domain than housing). Thus, a successful CSS represents a refinement that complements existing methods of assessing welfare and severity such as traditional score sheets or more complex scoring systems (*e.g.*, *Honess & Wolfensohn, 2010*; *Wolfensohn et al., 2015*; *Bugnon, Heimann & Thallmair, 2016*; *Ullmann et al., 2018*).

We tested our CSS concept using non-human primates in the context of neuroscience research as our example model organism and research environment. Non-human primates are particularly important animal research models in neuroscience research due to their highly developed cognitive abilities that enable them to learn new associations and perform complex sensory discrimination and motor tasks (*Roelfsema & Treue, 2014*). Our application of the CSS focused on an experimental testing welfare parameter: the location where cognitive testing took place. We offered three adult male rhesus macaques (*Macaca mulatta*) choices between two conditions: performing a basic experimental task in a typical neuroscience laboratory setup (*i.e.*, laboratory condition) *versus* performing the same experimental task in the monkeys' home environment (*i.e.*, cage conditions: either in the upper or lower cage of a testing compartment). The elements of these conditions differ in their states. In the laboratory condition, for example, the movement (*i.e.*, movement element) of the monkeys was constrained by a non-human primate chair to prevent equipment from being tampered with and ensure safe experimentation. In contrast to the laboratory condition, movement was less constrained in the cage conditions as the monkeys could move freely around in the cage. See Fig. 1 and Table 1 for an overview of the elements of these conditions. Given the differences between these conditions, the comparison between the laboratory and two different cage conditions offered a robust and practical test of our CSS concept *in situ* (*i.e.*, Choice-based Severity Assessment).

Accordingly, we expected that the monkeys would prefer to perform a basic experimental task in the cage conditions over the laboratory condition. To scale these conditions in relation to one another, we adjusted the number of fluid reward drops provided per correct trial of a basic experimental task and the type of reward in each condition. We tested our CSS concept further by offering the monkeys' choices between two experimental tasks varying in trial duration, where the amount of reward provided differed substantially (*i.e.*, CSS test). A CSS will determine which aspects of research have the highest impact on laboratory animal well-being from the animal's perspective. Our application of a CSS is one variation of how our welfare and severity assessment concept can be applied to welfare parameter scaling. Therefore, we provide guidelines to help implement our concept in other species and experimental settings.

## MATERIALS AND METHODS

Portions of this text were previously published as part of a thesis (*Cassidy, 2023*).

### Statement of ethics

Research with non-human primates represents a small but indispensable component of neuroscience research. The scientists in this study are aware of and committed to the great responsibility they have in ensuring the best possible science with the least possible harm to the animals (*Roelfsema & Treue, 2014*; *Treue & Lemon, 2022*).

This study and the procedures involving non-human primates were conducted according to the relevant national and international laws and guidelines, including the German Animal Protection Law, the European Union Directive 2010/63/EU on the Protection of Animals used for Scientific Purposes and the Society for Neuroscience Policies on the Use of Animals and Humans in Neuroscience Research. The procedures were approved by the responsible regional government office (Niedersaechsisches Landesamt fuer Verbraucherschutz und Lebensmittelsicherheit, LAVES) under the permit number 33.19-42502-04-18/2823. A research protocol was not pre-registered for this study. Following this study, the monkeys remained in the laboratory to be enrolled in different experiments.

### Study subjects and housing facility

We conducted the study on three adult male rhesus macaques (*Macaca mulatta*; 10, 11, and 19 years old at time of the Choice-based Severity Assessment) living at the German Primate Center, Goettingen, Germany. This number of study subjects is typical of behavioral neuroscience experiments, in line with the 3R principles of using the minimum number of animals necessary. These monkeys were purpose-bred in China (monkey D) or at the German Primate Center (monkey H, monkey E) and entered the Cognitive Neuroscience Laboratory between 3 to 5 years old. Importantly, the monkeys were not naïve to the conditions experienced in the study. All monkeys had extensive positive-reinforcement training (4 years or greater) to facilitate handling for husbandry and experimental purposes, particularly for cooperatively entering and sitting in a non-human primate chair for long periods of time (*Bliss-Moreau, Theil & Moadab, 2013*;

*Ponce et al., 2016*; *Mason et al., 2019*). Furthermore, all monkeys learned to perform a basic experimental task (at a proficiency of 80% or more of trials in a session) offered in a typical neuroscience research setup (*i.e.*, laboratory condition) and close to their home cage (*i.e.*, cage condition) prior to the study. After these training criteria were met, the selection of our study subjects was opportunistic and based on availability, *i.e.*, which monkeys were not involved in neuroscience experiments at the time of the study. More specific information about the study subjects can be found in Supp. Table 1 of the Supplemental Material.

All monkeys were socially housed in isosexual pairs, with visual and auditory contact to other macaque groups. The monkey housing consisted of two large compartments, which exceeded the size requirements for macaques set by EU directive 2010/63/EU. The indoor compartment was temperature regulated with a 12-h light/dark cycle (from 07:00 to 19:00) and was connected by a tunnel to a sheltered outdoor compartment where the monkeys could see outside and experience natural climate fluctuations (*e.g.*, light, temperature, wind). Both compartments were carpeted with wood shavings and furnished with environmental enrichment (*e.g.*, balls, cardboard) and a variety of perches (*e.g.*, raised platforms, ropes; also described in *Cassidy et al., 2021*). On days where the monkeys were not tested, they had access to monkey chow, fresh fruits and vegetables, and water *ad libitum*. On training and test days, monkey chow and dried foods were accessible *ad libitum*. Unlimited fluid was available while the animal engaged in the study's cognitive tasks, as is typical of neuroscience research laboratories (*i.e.*, controlled access to fluid; described in *Pfefferle et al., 2018*). Additionally, the monkeys were weighed each training and test day. Daily health monitoring of the monkeys was carried out by veterinarians, monkey facility staff, and researchers who all have specialized training for working with non-human primates.

## Experimental testing apparatuses

We used multiple cognitive testing systems (*i.e.*, eXperimental Behavioral Instruments: *Calapai et al., 2017*; *Berger et al., 2018*) to present condition stimuli during the CSS test and administer the cage condition tasks (*i.e.*, basic experimental task, delivery of 2 ml bolus). These standalone systems were developed within the laboratory (Cognitive Neuroscience Laboratory, German Primate Center) to facilitate cage-side cognitive task training and testing. In our study, the monkeys could engage with a task by using the touchscreen and sensors equipped to the cognitive testing systems. When needed, fluid reward was dispensed *via* a tube positioned about 45 cm in front of the touchscreen (30.4 cm by 22.7 cm; 60–75 Hz framerate). The positioning of the reward tube on these cognitive testing systems encourages monkeys to adopt stereotypical postures when engaging with cognitive tasks (*Calapai et al., 2017*). Multiple cognitive testing systems were mounted to a flexible testing compartment attached to each monkey's home cage, which could be divided into quadrants (approximately 80 cm by 75 cm by 90 cm) home cage. We programmed all cognitive tasks using MWorks (versions 0.8 to 0.10; https://mworks.github.io/). MWorks is an open-source C++-based software that allows for the design and implementation of real-time controlled behavioral tasks (*Calapai et al., 2017*).

## Choice-based severity assessment

Our main aim was to test the CSS concept through a Choice-based Severity Assessment. We developed an experimental setup (Fig. 2) to offer three adult male rhesus macaques a choice between performing a basic experimental task in the cage or laboratory condition. Choice testing was conducted in the testing compartments, where the choice between the conditions was presented using visual stimuli on a neutral cognitive testing system (*Calapai et al., 2017*; *Berger et al., 2018*) and the conditions were positioned on different quadrants of the testing compartment (Fig. 2). We found that this experimental setup limited the potential influence from the environment and/or experimenter best through a series of pilot experiments summarized in the section 'Supplementary experiments' of the Supplemental Material. A video demonstration of the Choice-based Severity Scale concept and our example of a Choice-based Severity Assessment can be found in the Supplemental Information.

### *Experimental conditions of the choice-based severity assessment*

In our typical neuroscience research setup (*i.e.*, laboratory condition), monkeys were transported in a non-human primate chair by a researcher to a small, darkened experimental room. This experimental room was equipped with devices typical of a visual neuroscience laboratory: computer monitor for the presentation of visual stimuli, various non-human primate chair attachments (*e.g.*, sensor response box, reward delivery tube), eye-tracker, and a fluid delivery system (peristaltic pump). The researcher could administer and control cognitive tasks from a control center located just outside the door of the experimental room. In the laboratory condition, the monkeys were seated approximately 57 cm away from the computer monitor (59.7 cm by 33.6 cm; 120 Hz framerate). In our study, monkeys could respond to the basic experimental task presented on the computer monitor *via* a proximity sensor (*i.e.*, 'sensor') and received fluid for correct trials *via* a reward tube attached to the non-human primate chair.

The cage conditions took place in different quadrants of the testing compartments next to the monkeys' home cage (Fig. 2). All monkeys were trained to voluntarily enter this compartment for training, testing, temporary separation, experimental, and veterinary procedures as necessary. Quadrants were separated by movable sliding panels, which could be opened to shift monkeys between compartment quadrants and secured when the monkeys were present for longer durations. In the cage conditions, the monkeys could move around without restraint and had visual, acoustic, and/or olfactory contact to pair mates and adjacent social groups. Each quadrant had the capability to be equipped with cognitive testing system so that the monkeys could engage with a cognitive task without direct oversight from a researcher. Due to the cage location and ability to freely move around, we expected that the monkeys may prefer to perform their basic experimental task in the cage conditions over the laboratory condition.

### *Choice-based severity assessment protocol*

To generate a practically applicable CSS, we developed a protocol (2 reference sessions + 1 choice session; Fig. 2) for the Choice-based Severity Assessment that allowed the monkeys

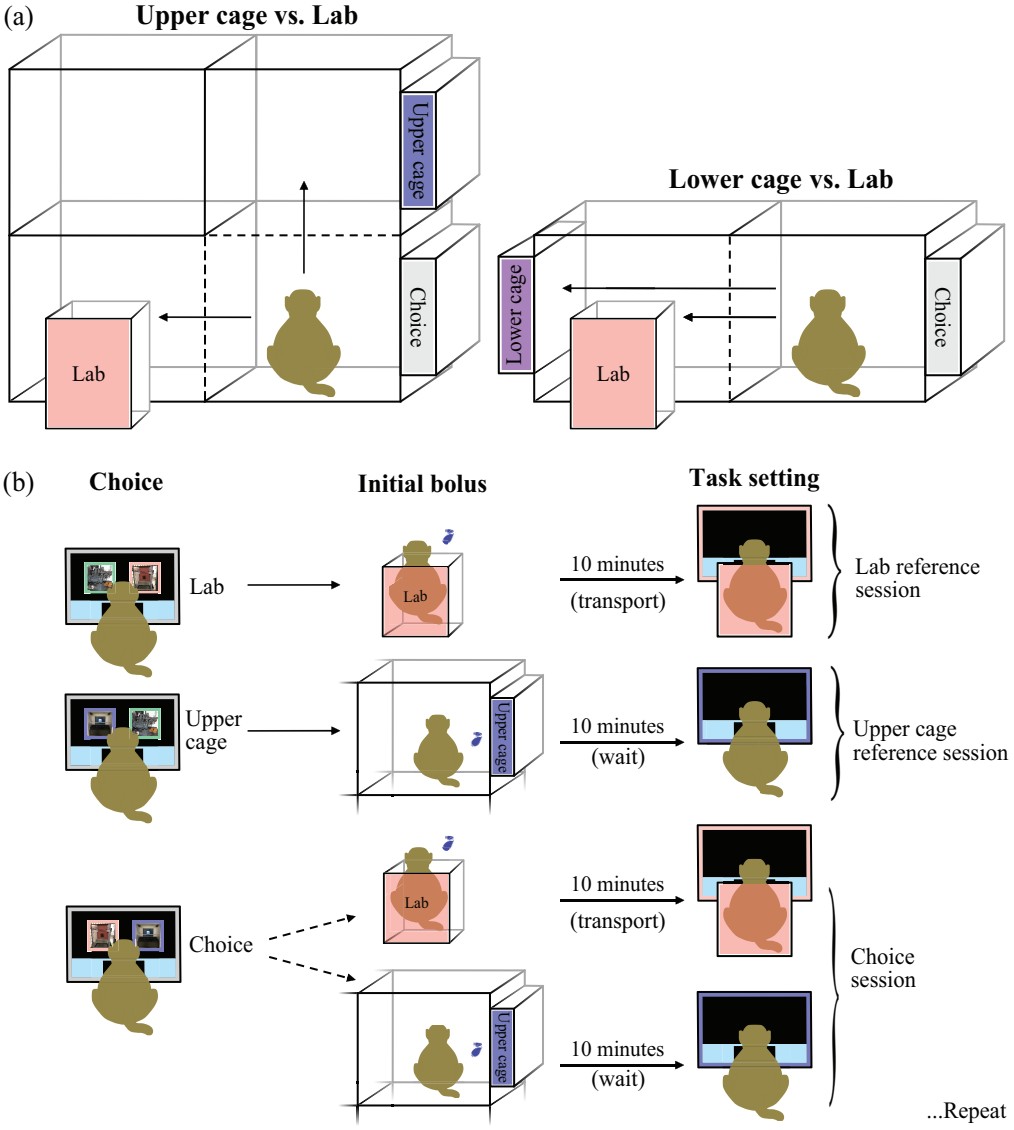

**Figure 2 Experimental setup and study design (Choice-based Severity Assessment protocol).** Laboratory is abbreviated as "Lab". (A) The grey box labeled 'Choice' indicates the location of the neutral cognitive testing system, where the monkeys made a choice between visual stimuli representing the cage and laboratory conditions (*i.e.*, condition stimuli). The cage conditions were positioned either on the upper right (*i.e.*, upper cage: blue box, representing a cognitive testing system, labeled 'Upper') or lower left quadrant (*i.e.*, lower cage: purple box, representing a cognitive testing system, labeled 'Lower') of a testing compartment adjacent to the monkeys' home cage. The laboratory condition was positioned on the lower left quadrant (pink non-human primate chair labeled 'Lab'). (B) Visual representation of the Choice-based Severity Assessment protocol, where the monkeys were given a reference trial for each condition prior to the choice between a cage condition and the laboratory condition. During reference sessions, a condition stimulus (in pink or blue) was presented simultaneously with a timeout stimulus (in green) that was unrewarded. In this example, the monkey is given two reference sessions and a choice session between performing a basic experimental task in the upper cage (blue) or laboratory condition (pink). Each session, the monkey selected a condition stimulus ('Choice'), followed by a small motivational reward ('Initial bolus') once it was seated in the non-human primate chair (representing the laboratory condition) or from the cage condition cognitive testing system. The basic experimental task was started once the monkey had been transported to the neuroscience setup (approximately 10 min) or

**Figure 2** (continued)
after 10 min on the cage condition cognitive testing system, matching the time course of the laboratory condition ('Task setting'). Then, the monkey could conduct as many trials as desired within 2 h before it was returned home. This protocol was repeated after two reference sessions and one choice session. All graphics and images were drawn or taken by Lauren Cassidy.

to experience the full consequences of each condition (*i.e.*, reference sessions) prior to choosing between the two conditions (*i.e.*, choice session). Monkeys were either given one reference or choice session a day, followed by the procedure of the corresponding condition. In each condition, the monkeys could work on a basic experimental task for as many trials as they desired within 2 h if they continued to engage in the task (there was a regulatory requirement to provide ample time for the monkeys to collect as much reward as they desired). The basic experimental task automatically stopped once it detected no engagement for a predefined duration (conclusion criteria was individualized), and the experimenter returned the monkeys to their home cage soon afterwards. All training and basic experimental task details are described in depth in the Supplemental Material.

To scale these conditions in relation to one another, we sought to influence the monkeys' choices by changing the reward contingencies of the basic experimental task itself. Here, we adjusted the number of fluid drops (approximately 0.3 ml each) provided per correct trial in each condition depending on the monkeys' choices until the combination of the conditions and their corresponding amount of reward is perceived as equal (*i.e.*, oscillating around a point of subjective equality). This adaptive approach is a popular method used in human psychophysics experiments to determine perceptual thresholds (*Leek, 2001*; *Kingdom & Prins, 2010*) and forms the basis of automated training protocols to shape complex behaviors in non-human primates (*Berger et al., 2018*; *Calapai et al., 2022*). At the beginning of testing, we set the difference in reward per trial between the two conditions to be large, where the number of drops of reward per trial in the laboratory condition was nine times larger than in the cage condition (laboratory: nine drops; cage: one drop). The monkeys' preferences were assessed after every three choice sessions (*i.e.*, bouts) and the reward per trial was adjusted so that the reward per trial of the preferred condition was reduced by two drops and the non-preferred condition increased by two drops (bounded by one and nine drops). For example, if the monkey exhibited a preference for the laboratory condition in the first three choice sessions, then the reward per trial for that condition would be reduced from nine to seven drops and the reward per trial for the cage condition would be increased from one to three drops. We concluded testing if the reward per trial difference was at the extremes (one and nine drops) and if the monkeys made the same choice for six consecutive sessions, irrespective if a bout was finished.

### Testing phases of the choice-based severity assessment

Through the Choice-based Severity Assessment, we tested two monkeys over three phases (a third monkey was tested only during phase 3), where we controlled the position of the

cage condition and adjusted the type of reward provided in the basic experimental task of each condition. Each monkey served as its own control by comparing individual preferences across all conditions (experimental unit: single animal). During the first phase, the cage condition (indicated by a cognitive testing system) was positioned on the upper right quadrant and the laboratory condition (indicated by a non-human primate chair) on the lower left quadrant so that the distance between a neutral cognitive testing system and each option was roughly the same (Fig. 2). To ensure that the monkeys made choices based on a preference for the condition instead a preferred quadrant of the testing compartment, we moved the cage condition to the same quadrant that the laboratory condition was positioned (lower quadrant to the left of the neutral cognitive testing system). Then we tested the monkeys again in a second phase of the experiment. The type of reward per trial was the same (grape juice) for all monkeys and conditions during the first and second phases. To test if the monkeys would change their preference due to the type of reward provided in each condition, we tested a third phase. During the third phase, water was received from the basic experimental task in the cage condition (preferred option during phase 2) and the monkeys' preferred juice was received from the same task in the laboratory condition (see the section 'Fluid preference test' in the Supplemental Material for more information).

### Procedure for the choice-based severity assessment

Each day the monkey was brought into the test compartment where the neutral cognitive testing system was mounted (Fig. 2). Once the monkey was seated in front of the neutral cognitive testing system, the experimenter remotely triggered the start button to appear. After the monkey touched the start button, two stimuli appeared for the monkey to choose between (reference session: condition stimulus and timeout stimulus; choice session: two different condition stimuli). If a condition stimulus was touched, the experimenter opened the corresponding compartment and the monkey received a small motivational reward (*i.e.*, 2 ml bolus of water) either by triggering the cage condition cognitive testing system (cage condition) or from the experimenter once seated in the non-human primate chair (laboratory condition). For laboratory condition choices, the experimenter then transported the monkey in the non-human primate chair to the neuroscience setup, attached the fluid reward system, and began the basic experimental task (approximately 10 min). For cage condition choices, the experimenter then removed the non-human primate chair and left the room for 10 min, to match the time course of the laboratory condition, before starting the basic experimental task. For both conditions, the monkey was returned to his home cage as soon as he stopped engaging in the task in either condition (see the 'Basic experimental task training for the Choice-based Severity Assessment' in the Supplemental Material for the conclusion criteria). The monkeys always made a choice and never chose the timeout stimulus during the reference sessions of the three experimental phases.

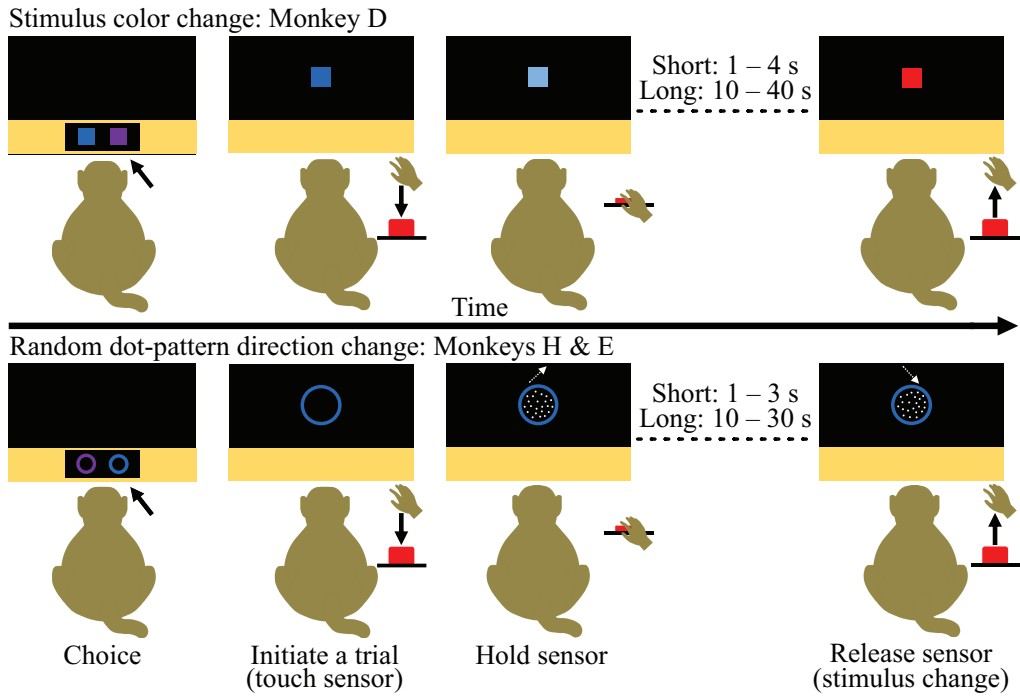

**Figure 3 Time courses of the tasks provided during the Choice-based Severity Scale test.** The monkeys indicated their choice by touching one of the task stimuli presented on the touchscreen of a cognitive testing system and were rewarded with 0.15 ml water to encourage engagement (first panel). The chosen stimulus appeared larger in the center of the touchscreen and blinked every 2.5 s until the monkey initiated a trial (second panel). Trials were initiated by the monkey touching and holding a proximity sensor (*i.e.*, 'sensor'; second panel). The stimulus either deluminated (stimulus color change task) or a random dot-pattern appeared moving in one direction (random dot-pattern direction change task) upon touch and the monkey had to hold the sensor until there was a second change in the stimulus (either another color change or change in the direction of the random dot-pattern; third panel). The duration of the hold depended on the monkeys' choice. Once the stimulus changed, the monkeys had to release the sensor within 2.5 s to receive the fluid reward associated with their choice (fourth panel). All graphics were drawn by Lauren Cassidy.

## Choice-based severity scale test

To test the scaling aspect of our CSS concept further, we conducted an additional experiment in which we offered the same three monkeys choices between experimental tasks varying in trial duration. We adapted the monkeys' basic experimental task to create two tasks that differed by a factor of 10 in the duration of how long the monkeys needed to hold a sensor until a stimulus change occurred (*i.e.*, different effort needed to complete each task) and the reward the monkeys received (Fig. 3). To determine if we could influence the monkeys' choices using reward, we provided 15 ml more reward for selecting and successfully completing the long hold task over the short hold task (the two tasks also differed in the type of fluid reward; long: preferred juice; short: water). To account for position biases (see the section 'Supplementary experiments' in the Supplemental Material for a pilot experiment), we set the position of the task stimuli to be deterministic, where the long hold task appeared in the opposite position as the last choice trial. Therefore, the task stimuli would alternate every trial if the monkey exclusively chose the long hold task.

Conversely, if the monkey chose the short hold task, the position of the stimuli would remain the same. Training for the CSS test is described in the section 'Preparation of the Choice-based Severity Scale test' in the Supplemental Material.

During the CSS test, the monkeys were given a choice on a trial-by-trial basis between the short and long hold task (Fig. 3). Each monkey was tested for 10 days, and their behavior was analyzed by individual to determine individual preferences (experimental unit: single animal).

## Statistical Analyses

We conducted statistical analyses to investigate the monkeys' training performance for the condition stimuli prior to the Choice-based Severity Assessment (model and results described in the section 'Model description and results of the condition stimuli training for the Choice-based Severity Assessment' in the Supplemental Material) and their choice behavior during the CSS test (model described below). Generally, we fit Generalized Linear Mixed Models (GLMMs), specified with a binomial distribution and a logit-link function, using a Bayesian framework *via* the 'brms' package (version 2.16.3: *Bürkner, 2017*) in R (version 4.1.2: *R Core Team, 2021*). brms calls Stan, a computational framework, to fit Bayesian models (*Bürkner, 2017*). No choice data were excluded from these analyses.

For all GLMMs, fixed effects included in each model did not correlate above 0.5 (using Spearman's correlation). We checked the distributions of model covariates and log transformed them when needed (*i.e.*, trial number). Covariates were also z-transformed to a mean of 0 and a standard deviation of 1 to provide more comparable estimates and aid the interpretation of any interactions (*Aiken, West & Reno, 1991*; *Schielzeth, 2010*). We used weakly informative priors to improve convergence, avoid overfitting, and to regularize parameter estimates (*McElreath, 2020*). Binomial models had priors for each intercept that were a normal distribution with a mean of 0 and a standard deviation of 1. The priors for the beta coefficients were also a normal distribution with a mean of 0 and a standard deviation of 0.5. The priors for the standard deviation of group level (random) effects an exponential distribution with scale parameter 1. The priors for correlations between random slopes were LKJ Cholesky priors with scale parameter 2.

Each model was run using four MCMC chains for 2,500 iterations, including 1,000 "warm-up" iterations for each chain, with convergence of the chains confirmed by there being no divergent transitions, all Rhat values were equal to 1.00, and visual inspection of the plotted chains. We also checked model performance by using the 'posterior predictive check' ('pp_check') function from the 'bayesplot' package (*Gabry & Mahr, 2022*). We report model estimates as the mean of the posterior distribution with 95% credible intervals (CI). To aid in the interpretation, we calculated the proportion of posterior samples that fell on the same side of 0 as the mean (Pr) to understand whether the fixed effects substantially influenced performance and choice behavior. The Pr ranges from 0.5 to 1.0, where a Pr of 1.0 indicates a strong effect of a predictor (either negative or positive) and a Pr of 0.5 indicates no effect of a predictor on the response.

To investigate whether the monkeys developed a preference for one task over the other during the CSS test, we fit three GLMMs (one per monkey) with the response variable as

whether the monkey chose the short or long hold task for each trial. We added session, the position of the monkeys' choice (left or right), and the amount of reward accumulated as fixed effects. We also included session as a random effect with all possible random slopes (*Schielzeth & Forstmeier, 2009*; *Barr et al., 2013*).

# RESULTS

## Applying a choice-based severity assessment

Through the Choice-based Severity Assessment protocol (see Choice-based Severity Assessment protocol) and experimental setup (Fig. 2), we offered adult male rhesus macaques a choice between performing a basic experimental task in a cage or laboratory condition to generate a CSS. We found evidence of inter-individual differences in condition preference and how the monkeys responded to changes in the reward contingencies. During the first two phases, where the position of the cage condition was controlled for (changed from the upper quadrant to the lower quadrant, where the laboratory condition was positioned), monkey H exhibited a strong preference for the cage condition (100% of six choice sessions each; Fig. 4). Notably, this preference occurred despite the reward per trial being largely in favor of the laboratory condition (Fig. 4). Once the reward per trial in the lower cage condition changed from grape juice to water (Fig. 4; laboratory condition reward remained grape juice), monkey H switched his preference to the laboratory condition (75% of 18 choice sessions). Monkey H's preference for the laboratory condition persisted despite the reward per trial increasing to become largely in favor of the cage condition (Fig. 4).

In contrast, monkey D exhibited an initial preference for the laboratory condition during the first bout of the first phase (upper cage *vs* laboratory condition, type of reward per trial was grape juice for both; Fig. 4). Further into choice testing, however, monkey D switched his preference to the upper cage condition, irrespective of the amount of reward per trial in each condition (upper cage condition was chosen in 75% of 16 choice sessions; Fig. 4). Monkey D also chose the lower cage condition during the second and third phase (100% of six choice sessions each), despite the location of the condition being controlled for, the type of reward per trial changing (cage: grape juice to water; laboratory: grape juice to banana juice), and amount of reward per trial being largely in favor of the laboratory condition (Fig. 4). These data suggest that during the first phase monkey D sampled the different conditions, then settled on selecting the cage condition exclusively at the end of this first phase and continued to do so during the next two phases.

We tested an additional monkey (monkey E) during the third phase of the Choice-based Severity Assessment. Monkey E exhibited the same preference as monkey D, where he exclusively chose the lower cage condition (100% of six choice sessions), despite the amount and type of reward per trial being largely in favor of the laboratory condition (see the section 'Results of the Choice-based Severity Assessment for the third monkey (E)' in the Supplemental Material).

It should be noted that in our neuroscience setup the monkeys could easily compensate for lower reward per trial (typically experienced in the cage conditions) by performing more trials. Accordingly, the monkeys performed more trials on average in the cage
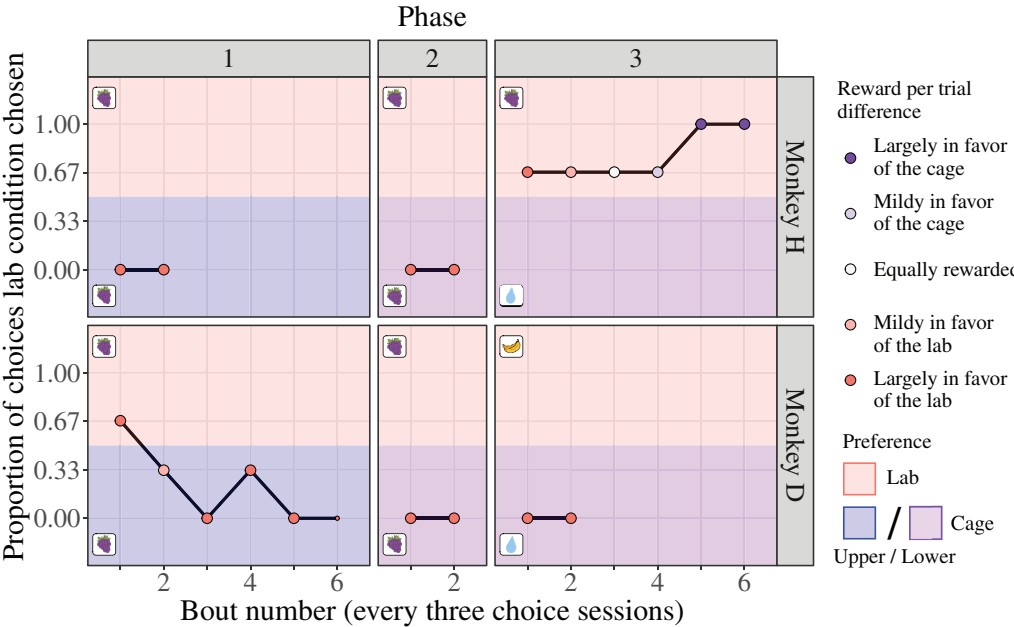

**Figure 4** **Results of the Choice-based Severity Assessment.** Laboratory is abbreviated as "lab". The data are separated by each phase for the two monkeys that were tested in all phases. Two references sessions (one per condition) preceded each choice session to remind the monkey of the consequences associated with each condition stimulus. One bout usually consisted of three consecutive choice sessions and their reference sessions, which took nine days to complete. Proportions were calculated for each bout (represented by point size, ranging from 1 to 3 choice sessions) to assess preference and adjust the reward per trial difference for the next bout accordingly. The reward per trial difference is indicated by point color. The type of reward used for each condition of each phase is indicated by the boxed picture on each panel. Over phases 1 and 2, grape juice (indicated by grapes) was delivered as a reward in each setting. During phase 3, the type of reward in the laboratory condition was changed to water (indicated by a drop of water) and the lower cage condition was changed to the monkey's preferred reward (monkey H: grape juice; monkey D: banana juice). A third monkey (monkey E) was only tested on the third phase and exhibited the same behavior as monkey D in phase 3 (see the section 'Results of the Choice-based Severity Assessment for the third monkey (E) in the Supplemental Material). The water drop graphics were drawn by Lauren Cassidy. The grape graphics were obtained from https://clipartspub.com and the banana graphic from https://www.clipartbest.com.

conditions than the laboratory condition (lower cage: 666 ± 342 trials; upper cage: 964 ± 443 trials; laboratory: 97 ± 59 trials) when the fluid reward type was the same. Furthermore, the monkeys spent a greater amount of time working in the cage conditions then in the laboratory condition on average when the fluid reward type was the same (lower cage: 97 ± 33 min; upper cage: 94 ± 28 min; laboratory: 26 ± 9 min).

## Choice-based severity scale test

The CSS test applied our CSS concept further by offering the monkeys choices between experimental tasks varying in trial duration. We found strong evidence that monkey H chose the long hold task more frequently overall, irrespective of the position of the task stimuli and session (Fig. 5; Table 2). In contrast, there was strong evidence that the position of the task stimuli influenced the choice behavior of monkey D, where the long hold task was chosen less frequently when positioned on the left of the touchscreen (Fig. 5; Table 2).

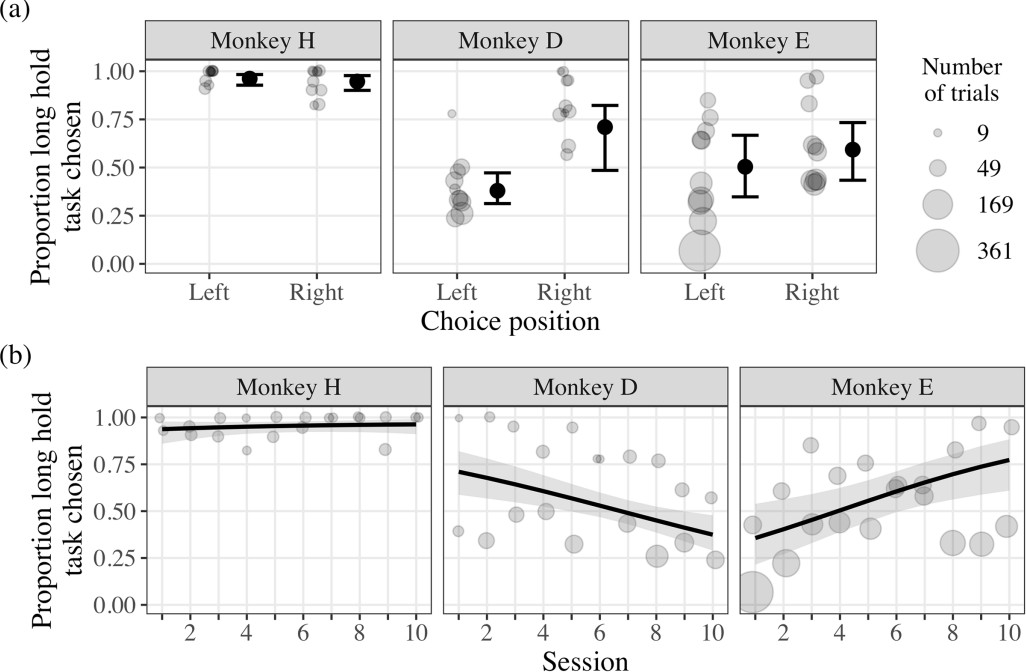

**Figure 5 Results of the Choice-based Severity Scale test.** The light grey points indicate the proportion of choices the long hold option was chosen over the total number for trials for each choice position, session, and monkey (range: 9 to 338 trials). In plot (A), the data are plotted by the position of the monkeys' choices, where the large black points indicate the model probability estimates and the black whiskers indicate the 95% credible intervals. In plot (B), the data are plotted by the proportion of trials the long hold task was chosen by session, where the black lines indicate the model probability estimates across sessions and the shaded gray areas indicate the 95% credible intervals.

Given that the position of the long hold task was deterministic due to our experience in previous pilot experiments (see the section 'Supplementary experiments' in the Supplemental Material), our results suggest that monkey D had a left side bias, causing the short hold stimulus to appear on the left repeatedly. However, monkey D had to interrupt this bias to choose the long hold option, suggesting that these choices were deliberately made. Additionally, there was moderate evidence that session influenced the choice behavior of monkey D, where he selected the long hold task less frequently as the number of sessions increased (Fig. 5; Table 2). There was little evidence that the choice behavior of monkey E was influenced by the position of the stimuli, but moderate evidence that he selected the long hold task more frequently as the number of sessions increased (Fig. 5; Table 2). Such behavior suggests that, with additional sessions, monkey E learned that the tradeoff for engaging with the long hold task was more favorable with additional sessions.

Monkey H was the most efficient (*i.e.*, least effort for most reward) and received 14.4 ml per trial on average across sessions, whereas monkey D and monkey E received 7.8 and 7.1 ml per trial respectively on average across sessions. While these descriptive statistics suggest that the strategies monkey D and E employed were not optimal, the monkeys were still able to receive over 400 ml reward per session on average by choosing the long hold

**Table 2 Model results for the Choice-based Severity test.** The binomial generalized linear mixed models for each monkey tested whether the short or long hold task was chosen.

| Monkey | Variable | Estimate | SD | Lower CI | Upper CI | Pr |
|---|---|---|---|---|---|---|
| H | Intercept | 3.29 | 0.38 | 2.55 | 4.05 | 1.00 |
| | Choice position (right)[a] | −0.34 | 0.39 | −1.09 | 0.42 | 0.81 |
| | Session | 0.22 | 0.28 | −0.32 | 0.80 | 0.78 |
| | Trial number | −0.21 | 0.25 | −0.70 | 0.27 | 0.80 |
| D | Intercept | −0.49 | 0.17 | −0.79 | −0.11 | 0.99 |
| | Choice position (right)[a] | 1.42 | 0.42 | 0.43 | 2.04 | 1.00 |
| | Session | −0.31 | 0.14 | −0.59 | −0.03 | 0.98 |
| | Trial number | −0.05 | 0.14 | −0.34 | 0.20 | 0.64 |
| E | Intercept | 0.02 | 0.33 | −0.63 | 0.70 | 0.51 |
| | Choice position (right)[a] | 0.37 | 0.39 | −0.43 | 1.09 | 0.83 |
| | Session | 0.59 | 0.27 | −0.03 | 1.05 | 0.97 |
| | Trial number | 0.18 | 0.16 | −0.15 | 0.48 | 0.88 |

Notes:
Estimate, slope of the predictor; SD, standard deviation of the estimate; CI, 95% credible interval; Pr, proportion of the posterior samples that fall on the same side of 0 as the mean.
[a] Left was the reference level for choice position.

option occasionally and engaging in more trials. Thus, it may have not been necessary for the monkeys to exclusively choose the long hold option throughout the session.

## DISCUSSION

Objective assessment of laboratory animal welfare is crucial not only for ensuring that high standards of animal welfare are maintained, but for the validity and quality of the scientific experiments they are involved in *Poole (2007)*, *Jennings & Prescott (2009)*. To address the issues of ranking and scoring animal welfare parameters, we proposed and tested our Choice-based Severity Scale (CSS) concept (*i.e.*, using reward and choice-based preference testing to determine the utility of experimental testing conditions) by giving adult male rhesus macaques a series of choices to perform a basic experimental task close to their home cage (cage condition) or in a laboratory environment (laboratory condition). The data we collected consistently support the validity of the CSS concept, where we find distinct preferences for the conditions that we provided the monkeys and that these preferences can be influenced by changes in reward contingencies. During the Choice-based Severity Assessment, we limit the potential influence from the experimenter and/or environment on the monkeys' choices by providing the monkeys choices between visual stimuli associated with the conditions (*i.e.*, condition stimuli) on a neutral cognitive testing system. We provide guidelines in the Supplemental Material (see the section 'Guidelines for Choice-based Severity Assessments in animals') to highlight several points (*e.g.*, training, experimental setup, prior experience) to consider during the design of Choice-based Severity Assessments. Collectively, we believe that our study provides a basis for expanding and adapting the CSS concept to other species and other conditions than those we have explored in this study.

A core tenant of a Choice-based Severity Assessment is that it is applied individually. In support, the individual monkeys' choice behavior during our Choice-based Severity Assessment indicates that the CSS is indeed sensitive to inter-individual differences. During our Choice-based Severity Assessment, one monkey switched to the less preferable condition (*i.e.*, laboratory condition) given a large enough reward difference (juice instead water) in favor of that condition. Interestingly, the same monkey responded the strongest to the difference in reward amount per trial during the test of the CSS. While this behavior contrasts that of the others, it highlights that these monkeys may have different point of subjective equality for the costs and benefits associated with the conditions we tested (upper cage *vs.* laboratory, lower cage *vs.* laboratory). In other words, there may have not been enough incentive for the other two monkeys to select the less desirable condition due to the flexible time window to work in each setting and/or they may have not noticed the changes in reward contingencies (discussed in more detail later). Moreover, these monkeys were not naïve to the conditions of this study, so the choice behavior of these animals is not due to one of the conditions being novel (*Dawkins, 1977*; *Habedank, Kahnau & Lewejohann, 2021*; see the section 'Guidelines for Choice-based Severity Assessments in animals' in the Supplemental Material for further discussion). It is well known that individuals respond differently to their internal and external environments as aspects of their life histories differ (*e.g.*, species, age, sex, personality: *Sloan Wilson et al., 1994*; *Izzo, Bashaw & Campbell, 2011*; *Coleman, 2012*; *Palmer, Oppler & Graham, 2022*). Such differences are important to consider when designing the ranking and scaling of welfare parameters as animals do not perceive and experience welfare conditions in the same way. The CSS represents a severity assessment tool that matches this requirement in that it emphasizes the individual's experienced severity, rather than an across-animal magnitude of severity of a given condition as assessed using traditional approaches.

There are several explanations for why the amount of reward did not influence choice behavior during the Choice-based Severity Assessment. Given the regulatory requirement to provide ample time to collect as much reward as desired, the additional reward per trial might not have been enough incentive to choose the laboratory condition. Even though the laboratory condition is the most efficient way to gain fluid reward and return to the home enclosure earlier, the monkeys could easily compensate for this by performing more trials in the cage condition. This interpretation also stresses the importance of animals being in a similar state of satiety when conducting a series of choice tests involving food or fluid reward as the scaling metric, so that choice outcomes are comparable across choice sessions and not difficult to interpret. Alternatively, detection of reward contingency changes may have been hindered by the 10-min delay between the selection of a condition stimulus and its corresponding consequences. This 10-min delay was necessary to transport the monkeys to the location of the laboratory condition and was matched with a waiting period 10-min in the time course of the cage condition. Within these delay periods, multiple distracting events could occur (*e.g.*, transport to laboratory, social group

interactions) that may have made the formation of an association between each stimulus and its outcome more challenging.

Our CSS testing data show that all three monkeys engaged in the long-hold task when the reward per trial was substantially higher than the short-hold task. These data support the core approach of the CSS that preference between two conditions can be reversed using reward amount. Thus, reward amount can be used a common unit to scale conditions across different parameters and domains in a comparable and animal-centric way. Given that we were able to reverse preference using reward amount for the CSS test and not the Choice-based Severity Assessment, differences in reward amount may be easier for animals to detect when the delay between a choice and its consequences is short (*e.g.*, delay was 40 s for the CSS test *vs.* 10 min for Choice-based Severity Assessment).

We recognize that giving the monkeys choices between the complex, full-scale experimental conditions in our study was time intensive. But as the CSS protocol closely reflected the actual procedures of the laboratory and cage conditions, we could build an accurate picture of how these conditions were experienced by the monkeys due to their choice behavior in the Choice-based Severity Assessment. Other conditions may not be as time intensive to determine animal preferences because visual stimuli may not be needed to represent each condition, which necessitate training sessions to remind the animal of the consequences of each condition stimulus. For example, offering choices between different types of bedding or enrichment devices would not require the items to be associated with species relevant stimuli because the items themselves could be offered simultaneously.

Importantly, how we incorporated the CSS concept into our Choice-based Severity Assessment is just one application in an example species and experimental setting. What conditions are presented and how the preference testing is carried out can be adapted to the species and experimental setting. Preference tests in mice, for example, often involve choices between different arms of mazes or conditioned compartments (reviewed in *Habedank et al., 2018*). Similarly, the reward to determine condition utility does not need to be imbedded within an experimental task. In the previous bedding and enrichment device example, reward could simply be provided adjacent to each condition.

The CSS concept has the capability to shed light into the current perspective of animals that other welfare and severity assessments have yet to tap into. However, there are limitations in the application of the CSS, particularly regarding internal physiological welfare parameters. Asking animals whether they prefer different physiological states, such as being thin or slightly overweight, is difficult to capture. Furthermore, there are conditions that are more abstract or that animals will likely never select, irrespective of how much reward is provided (*e.g.*, participating in a surgery). Thus, we stress that the CSS is relative (not absolute), based on those conditions that can be offered and that the animal knows what it is choosing between. We emphasize that the CSS is best used for assessing fundamental conditions the animals experience, as it is not suitable for a day-to-day routine severity assessment. Furthermore, as the CSS is dependent on the individual, a certain magnitude of preference (*i.e.*, reward differences), for example, cannot be

categorized as low or high in severity. As recent experience can shape perspective, we recommend applying a Choice-based Severity Assessment at regular intervals and/or during periods of distinct changes to experimental and husbandry practices as the perceived severity of these practices may change. Hence, the CSS concept complements the existing methods of welfare and severity assessment available to the stakeholders working directly with laboratory animals.

More development is needed to integrate our CSS concept into the current system of animal welfare and severity assessments. Naturally, testing more individuals is a good first step forward given the benefit of its animal-centric approach. Further validation by other individual-based welfare parameters such as physiology (*e.g.*, heart rate variability: *von Borell et al., 2007*), stress hormones (*e.g.*, *Pfefferle et al., 2018*), blood values (*e.g.*, *Wegener et al., 2021*), and behavior (*e.g.*, abnormal: *Gottlieb, Capitanio & McCowan, 2013*) is also warranted. Another interesting comparison would be to offer a condition that is putatively more positive in valence. In our laboratory, performing the basic experimental task in the home cage itself, where the monkeys have full visual access to conspecifics and can engage in other behaviors like foraging, is an alternative, putatively more positive, condition that could be compared. Conditions should be associated with species-relevant stimuli and a CSS protocol can be created to accommodate such conditions (*Kahnau et al., 2020*). For example, different compartments can be associated with different conditions (*e.g.*, conditioned place preference tests comparing, *e.g.*, food and an aversive procedure: *Millot et al., 2014*; social partners: *Panksepp & Lahvis, 2007*; analgesic drugs: *Roughan et al., 2014*) and offered simultaneously to animals to determine preferences. Lastly, expanding the CSS concept to test other species and other conditions warrants exploration.

## CONCLUSIONS

Historically, animal welfare science has shied away from recognizing animals' subjective experiences as meaningful to their welfare but interest in linking the two topics has grown in the last few decades (*Marchant-Forde, 2015*). The CSS concept that we propose here has fundamental benefits for making welfare and severity assessments less anthropocentric and more animal-centric by shifting the perspective of laboratory animals into the central focus. To our knowledge, our study is the first to offer laboratory animals choices between experimental procedures. In summary, the CSS is a powerful tool that can help shape the refinement of husbandry and research practices (*Schapiro & Lambeth, 2007*), and thus strengthen the validity and quality of scientific research.

## ACKNOWLEDGEMENTS

We thank the German Primate Center animal care and veterinary staff for taking care of the monkeys. We also thank the technicians of the Cognitive Neuroscience Laboratory for their help training the monkeys, Roger Mundry for his statistical help, and Ralf R. Brockhausen for task programming advice.

### Funding

Grants were awarded to Stefan Treue, Alexander Gail, and Dana Pfefferle from the German Research Foundation Research unit 2591 "Severity assessment in animal-based research" supported this study (https://severity-assessment.de; grants TR 447/5-1/2, GA 1475/6-1/2, and PF 659/5-2). Dana Pfefferle also received funding through a grant from the Leibniz ScienceCampus Primate Cognition (https://www.primate-cognition.eu/en/funding-measures/audacity-funds.html; grant LSC-2017-01SF). The funders had no role in study design, data collection and analysis, decision to publish, or preparation of the manuscript.

### Grant Disclosures

The following grant information was disclosed by the authors:
German Research Foundation Research Unit 2591: TR 447/5-1/2, GA1475/6-1/2, PF 659/5-2.
Leibniz ScienceCampus Primate Cognition: LSC-2017-01SF.

### Competing Interests

The authors declare that they have no competing interests.

### Author Contributions

- Lauren Cassidy conceived and designed the experiments, performed the experiments, analyzed the data, prepared figures and/or tables, authored or reviewed drafts of the article, and approved the final draft.
- Stefan Treue conceived and designed the experiments, authored or reviewed drafts of the article, and approved the final draft.
- Alexander Gail conceived and designed the experiments, authored or reviewed drafts of the article, and approved the final draft.
- Dana Pfefferle conceived and designed the experiments, authored or reviewed drafts of the article, and approved the final draft.

### Animal Ethics

The following information was supplied relating to ethical approvals (*i.e.*, approving body and any reference numbers):

The responsible authority for authorizing animal experimentation, Niedersaechsisches Landesamt fuer Verbraucherschutz und Lebensmittelsicherheit (LAVES), approved the animal experimentation application and procedures for research in the Cognitive Neuroscience Laboratory (permit number: 33.19-42502-04-18/2823).

### Data Availability

The data and analysis code are available in the Supplemental Files and at Goettingen Research Online: Treue, Stefan; Cassidy, Lauren; Gail, Alexander; Pfefferle, Dana, 2024, "Supplemental_files_V1", https://doi.org/10.25625/GRSAXC, GRO.data, V1.

## Supplemental Information

Supplemental information for this article can be found online at http://dx.doi.org/10.7717/peerj.17300#supplemental-information.

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
