# Peer review of "Choice-based severity scale (CSS): assessing the relative severity of procedures from a laboratory animal’s perspective"

_PeerJ, doi:10.7717/peerj.17300_

## Round 0.1 · original submission · Minor Revisions

I was fortunate to receive three detailed reviews from experts in this area. All three experts were enthused about the potential for this work, with Reviewer 1 being particularly excited about the novelty and importance of your approach. I share enthusiasm for this line of study and note that choice-based protocols are consistent with consent-based training for domestic dogs and other species in human care as well. Some of that literature (to the extent that it exists) could be relevant here. We all found the paper to be well written. Despite our shared enthusiasm for your approach, the reviewers caution you to be more transparent in terms of the limitations of the current data (based on such a small sample) and more forthcoming in terms of the expected utility of this approach and its role alongside other more established methods for assessing severity. All of the reviewers have provided very helpful feedback on how to improve the MS, which I hope you will follow. I have a few minor comments of my own:

The abstract should indicate how many subjects were tested (on line 22). How were these particular monkeys selected for participation in this study?

The idea of measuring how much an animal needs to be “paid” to do one task over another sounds like the idea of measuring how much effort an animal is willing to engage in for access to a given task or objects. Marian Stamp Dawkins wrote about this differential effort model. It might be useful to reference these paradigms as well.

Dawkins, M. S. (1990). From an animal's point of view: motivation, fitness, and animal welfare. Behavioral and brain sciences, 13(1), 1-9.

Reviewer 1 ·

Basic reporting

This manuscript is written clearly, is well organized and easy for the reader to follow. The figures and tables are easy to interpret, and they add to the value of the information contained in the written portions of the manuscript. The literature review is comprehensive and appropriate. I thoroughly enjoyed reading the paper and learned a lot from it.

I have a few, minor suggestions for improvement. (1) The title indicates the work relates to “laboratory animals” but the experiment involved three rhesus macaques. Although the work certainly has implications for other laboratory animals, I believe it would be more appropriate to reduce the scope indicated in the title to “rhesus macaques”. (2) In the abstract, please define “severity” as you do in the body of the paper, so the abstract will be more understandable to those not familiar with that terminology. (3) On line 92, can you describe why the papers cited about acupuncture and laser treatments are the animals “determining their own medical treatment”? If this is simply because they could have refused by refusing to cooperate with the positive reinforcement training, please state this.

Experimental design

The research questions were defined clearly, and the importance of these questions were set up well in the Introduction. The experimental conditions were explained well and were quite clever. For example, I think the use of extended trial durations and the manipulation of the reward magnitudes to allow scaling of the conditions was an excellent way to operationalize this. The care and treatment of the nonhuman primates met ethical standards. The details provided in the supplemental materials were very helpful and were clearly stated. The video was similarly very useful.

I have a few, minor suggestions for improvement. (1) Line 166. please include that the animals were under conditions of fluid restriction, as was described. (2) Line 287. Can you please describe the reasoning behind offering an immediate reward for one option, while the other leads to a delay (after being seated in the chair) prior to being reinforced. There were probably pros and cons considered in trying to match the reinforcement schedule across the two conditions, and it would be helpful to the reader to understand those considerations. (3) Please use the full word, laboratory, rather than the abbreviation “lab” throughout the paper. (4) Line 486 please insert “for” after “compensate.” (5) In the Figure 1a legend, please insert “be” after “may”.

Validity of the findings

The data have been provided and the statistical analyses are fully described and are appropriate. The conclusions about finding distinct individual preferences and illustrating how changing reward contingencies changed preferences are appropriate. The point made (around line 485) about it being easier and faster to study some other choices (such as different types of enrichment) is very important as the practical application (or lack of) of this approach will be a criticism as to how this could be used in practice is discussed. While the study only includes three subjects and therefore should be replicated with other subject groups, other species, and with other choices, the authors do a good job of describing the value of their work. As this study is the first, I know of to offer research nonhuman primates choices between how they will be involved in research procedures, and those choices are used to then scale the impact on welfare, this is a very important study and should be published.

Additional comments

Wonderful job bringing this important approach to the study of nonhuman primate welfare!

Reviewer 2 ·

Basic reporting

No comment

Experimental design

No comment

Validity of the findings

No comment

Additional comments

The laboratory and experimental work involving non-human primates draw particular attention from local authorities and the general public. Developing and assessing objective criteria for evaluating the severity of laboratory and experimental procedures is crucial to demonstrate their appropriateness. The authors propose a method for objectively assessing the severity of a given laboratory or experimental procedure by offering the monkey a choice between two alternative conditions and measuring the difference in reward as an indicator of their preference.

They test their "choice-based severity assessment" by presenting monkeys with two alternatives: entering the primate chair and performing a behavioural task in the laboratory versus performing the task in their compartment. The authors argue that the difference in reward required to make the monkey choose the lab option over the compartment option may serve as a measure to assess the severity of the laboratory procedure. The authors provide behavioural data from three monkeys that were given choices between two of altogether three different conditions (two cage conditions and one lab condition). The results for the individual monkeys are quite varied and can, in my opinion, be interpreted in different ways.

I admit that the idea of a choice-based, animal-centered approach to assess severity has some appeal, yet I see a number of problematic issues, both conceptually and in the way they are presented in the paper. Much of my criticism depends on the way the authors introduce their idea and how they advertise it. I think their concept must be discussed and put into context much more carefully. It remains unclear what the authors have in mind as the purpose of their choice-based severity assessment. Is it suggested to replace established and well-accepted forms of severity assessment? Is it proposed to serve as a source of information for authorities and ethics committees when applying for animal research permissions, or is it thought of as an in-house technique to assess protocols and lab routines? In the current form of the report, this is very unclear, and the way the authors advertise their idea over other established assessment techniques does not help to put their new technique into place.

Major issues:

1) The introduction contrasts their approach of an "animal-based" severity assessment with an "anthropocentric" assessment. The authors emphasize that anthropocentric assessment is prone to subjective biases, etc.

The way they introduce their system seems to suggest it may constitute the "better" system. I perceive this as an overstatement already in the introduction. As I said, I see the appeal. But I do believe, as I will argue in more detail below, that it can only serve as an additional system (if at all): The traditional way of assessing severity in the 3R framework is to use score sheets for registering a variety of parameters. These may be behavioural or physiological parameters, outer appearance, or other parameters that are supposed to be informative about the severity of the experimental condition and how an individual is affected by them. Score sheets were introduced as a way to gain objectivity, and they usually do a great job in this regard. The way the authors advertise their idea of a choice-based system seems to discredit this system. But how can a variety of parameters from different categories be adequately replaced by a system that collapses all these parameters into a single, reward difference-based system that is hard to interpret because it is prone to animal-based biases (see below)? If I got this wrong (i.e., if the authors do not suggest replacing the traditional scoring with their choice-based system but to use CSS as an additional way to assess severity), they need to be more careful and more explicit in this regard in their introduction.

2) The authors advertise their system as an animal-based severity assessment

Primarily, severity needs to be assessed when applying for permission from local authorities and/or ethics committees to conduct research, and during the research to evaluate all experimental procedures. Severity must be evaluated as mild, moderate, or severe. The authors criticise the traditional scoring system as anthropocentric and subjective. Yet, a difference in reward may indicate a preference of the monkey to choose one option over another, but it does not provide information about severity in terms of the three categories usually used to assess experimental procedures. Interpreting a difference in reward is entirely based on human judgments as well. In this regard, the suggested system is not different from the traditional scoring system at all: Does a difference of 50% reward to make option 2 more favourable indicate mild severity? Or moderate? Does it need to be 200% to be severe? The paper does not provide any guidelines on this or even discuss this point. Therefore, the argument of being "animal-based" and "objective" is not at all convincing.

Monkey H in their report may serve as an example: The monkey chose the more severe lab condition over the cage condition when it was offered water (if this is accurate—the authors report this in their results but indicate the opposite in their discussion). Most people would judge water as being less attractive than fruit juice, indicating that monkey H had an individual bias more in favour of the seemingly less attractive reward. Does this indicate the lab condition is less severe than the cage condition? According to table 1 of the paper, it is not. So, how to interpret this in terms of an objective severity assessment? I don't think this interpretation is very straightforward, and it is hard to believe it to be a more objective assessment than considering the behaviour of the monkey, its outer appearance, or other physiological parameters that were listed in the Discussion. Basically, even if the monkey preferred juice over water (opposite to what the authors report in Results), it is still possible that another monkey would choose water just because it has an individual bias. Moreover, even if the water/fruit juice preference of an individual is tested on beforehand, others studies have shown that such a preference may changes over time.

3) The authors suggest to assess severity as the reward difference between two conditions.

n order to use a difference in reward to assess the severity of condition 1 over condition 2, the monkey needs to be familiar with both conditions. In the case of the authors' study, all monkeys were familiar with the lab conditions. As such, determining the reward may be informative for a retrospective severity assessment, but since it is strongly dependent on the individual's preferences and presumably other individual parameters, it cannot be informative for a prospective severity assessment. Moreover, because the severity of a procedure depends on the familiarity of the individual with the procedure (particularly in NHP research, e.g., primate chair training is known to be more stressful at the beginning of the training than after a short while, as measured by cortisol concentrations), it can hardly be informative during the course of training, as long as the training is not completed. Therefore, at what time is the CSS informative about the severity, and for how long? The traditional scoring can be performed regularly, and it provides a continuous assessment option, while it is hard to see how the CSS might be used for this. The authors do not discuss this point.

4) The authors state that choice-based severity assessment is objective

The reward difference can only be informative (if at all) about the subjective difference of the two rewards offered. If condition 2 (e.g., the lab condition) is chosen over condition 1 (e.g., the cage condition), it provides information about the attractiveness of the rewards under the entirety of all husbandry and lab conditions the monkeys are subjected to, rather than the absolute severity of the condition. For example, if the monkey is only fed primate chow and has never access to fruits, offering a fruit juice might be much more incentive for this individual than for another having a diet providing regular access to fruits. The animal-based outcome will be informative about the incentive of the reward it is offered for the lab condition, but it is very unclear to what extent it is informative about the objective severity of the lab condition. If this was true (the authors neither tested nor discussed this point), a simple deprivation of the monkey during husbandry times (providing no fruits, or providing limited access to water) is very likely to bias the choice towards the seemingly more severe experimental condition. If the CSS is taken literally and is used to provide a severity assessment, the outcome would be a more moderate assessment under the deprived condition as compared to a less deprived condition, just because the monkey is more likely to chose the lab condition? Is this intended?

5) The study was performed on N=3 animals

The CSS is suggested as a protocol for severity assessment. As such, it should be easy to apply and should not require additional procedures that may induce additional burden, and it should provide reliable information. I wonder why the study has been performed on just three animals and during a few sessions only. It would have been informative to learn whether the choices of the animals are consistent during different periods of laboratory work. It would have been informative to learn about the choices of more animals trained to the task (or do the three monkeys constitute the entire group of animals trained on a lab condition?). Making the CSS a routine assessment to be applied in other labs would also require offering alternatives for introducing a choice-based system: Not all labs can easily introduce the cage conditions the authors were using, e.g., due to non-standard hardware required for it. This topic is important as authorities may use the suggestions of the paper to require information of this kind. If this information is hard to provide, or if the results are difficult to interpret, or if there is not much knowledge about the consistency over time, or dependency on other, seemingly less important parameters (e.g., the monkeys in the authors' study were weighed each day, i.e. they entered the primate chair anyway — would they choose differently if they were asked for their choice when already placed in the primate chair? Or would the reward difference just be smaller, as expected because the absolute difference in severity is smaller?), this sort of open questions causes uncertainty for an "objective" severity assessment and may also cause a lot of trouble for dealing with authorities.


Taking together, I see the appeal of an animal- and choice-based assessment, but I do not believe, and I am not convinced, that it provides any alternative to the currently used scoring system. As such, I think the idea of CSS needs a much more careful introduction and discussion. It needs to be put into place, presumably as an additional method to assess preferences of individual animals, with the registered data serving as examples for the choices and biases of individual animals. Personally, I think that the data mainly show this: Different animals experience one and the same situation differently, they come up with different choices, presumably due to (at least additional) soft factors which are not easy to catch.

Reviewer 3 ·

Basic reporting

This manuscript is well-written. The described background is clear and relevant, aligning with the further study and the obtained data. The most relevant references in this field of research have been included. No further comments on the basic reporting.

Experimental design

The relevant research fits within the scope of PeerJ. The research question is clear and well-described, focusing on an important aspect related to animal welfare for non-human primates, particularly those included in neuroscience research. The experimental design is clearly outlined and logical in the context of the research question. No further comments.

Validity of the findings

All relevant data are included in the text or supplementary data and figures. The conclusion is clear and aligned with the research question.

However, while the aspects of CCS are evident, the conclusion is based on only 3 animals. It is not entirely clear what the background of these animals is concerning how long they have been trained in this type of research or whether they are naive and untrained animals when they entered the study. Particularly, the duration for which naive animals should be trained to achieve the results described here should be clearly stated and discussed.
Although CCS can indeed be an important tool for reducing anthropocentric thinking regarding animal welfare and choices, and it appears particularly relevant in NHP neuroscience research, the general conclusion that CCS can be a significant tool in refining husbandry and research practices in animals in biomedical research is too optimistic. This should be better discussed and substantiated, explaining why it could be considered a general tool. This also relates to the title, which suggests a general applicability while the focus is mainly on NHP in neuroscience research.

Additional comments

This is an interesting manuscript with potentially promising implications for determining animal preferences regarding the performance of specific tests in scientific research. Providing animals with the (limited) opportunity to express their preferences on how they want to conduct certain tests is certainly novel and very intriguing. Granting choices to animals adds certainly a level of refinement. The question is how widely applicable this is, both in terms of the type of research, in this case neuroscience, and in terms of the species, in this case non-human primates. The authors should discuss this in more detail. If there are major limitations, the title should be adapted to reflect a more specific scope

---

## Round 0.2 · accepted · Accept

One of the previous reviewers was available to review your revision. Given that the previous recommended revisions were generally minor, I have proceeded with this favorable review and a careful read of the manuscript myself. I thank you for submitting such a clearly-written and important paper. I have no additional requests and am happy to recommend the paper for publication.

Reviewer 2 ·

Basic reporting

The authors submitted a thorough revision of their paper on choice-based severity assessment. In my review of their original submission, I have indicated a number of concerns, all of them mainly caused by (or due to) the fact that the purpose and goal of the CSS method were not sufficiently well explained to exclude misunderstandings regarding its areas of application and significance. In the revised version, the authors now clearly indicate that the CSS needs to be thought of as a complementary method, and the method is now put well into place with regard to other, established scoring methods. Therefore, my main concern has been addressed and I thank the authors for clarifying this issue.

Apart from this, I agree with the other Reviewers that the paper is well written and the scientific methods are sound and well described, and Figures and Supplementary Materials are informative. I recommend and support accepting the ms for publication and i think it will provide a valuable contribution to the discussion of NHP welfare in laboratory settings.

I recognised a few typos (lines refer to red line version):
- L146: "was" should read "were"
- L256: "perisaltic" should read "peristaltic"
- L553: "the these" should read "these"
- L566: "a mazes" should read "mazes"
- L580: "reward different" should better read "reward difference"?

Experimental design

The experimental design meets the scientific standards, reporting of methods and results is clear and complete.

Validity of the findings

The authors test their CSS concept in three monkeys. The results show that different monkeys may have different choices and underline the animal-centered approach for assessing the individual severity of a laboratory procedure. Results are discussed careful and they are put in context.

Additional comments

Dear Editor - is the explanation on the Potential Conflict part of the